# Autoimmune Thyroiditis and Vitamin D

**DOI:** 10.3390/ijms25063154

**Published:** 2024-03-09

**Authors:** Teodoro Durá-Travé, Fidel Gallinas-Victoriano

**Affiliations:** 1Department of Pediatrics, School of Medicine, University of Navarra, 431008 Pamplona, Spain; 2Navarrabiomed (Biomedical Research Center), 31008 Pamplona, Spain; fivictoriano@hotmail.com; 3Department of Pediatrics, Navarra Hospital Complex, 31008 Pamplona, Spain

**Keywords:** anti-thyroglobulin antibodies, anti-thyroid peroxidase, autoimmunity, autoimmune thyroiditis, Hashimoto thyroiditis, immune cells, vitamin D, vitamin D deficiency, vitamin D supplementation

## Abstract

Hashimoto’s thyroiditis (HT) is marked by self-tissue destruction as a consequence of an alteration in the adaptive immune response that entails the evasion of immune regulation. Vitamin D carries out an immunomodulatory role that appears to promote immune tolerance. The aim of this study is to elaborate a narrative review of the relationship between vitamin D status and HT and the role of vitamin D supplementation in reducing HT risk by modulating the immune system. There is extensive literature confirming that vitamin D levels are significantly lower in HT patients compared to healthy people. On the other hand, after the supplementation with cholecalciferol in patients with HT and vitamin D deficiency, thyroid autoantibody titers decreased significantly. Further knowledge of the beneficial effects of vitamin D in the prevention and treatment of autoimmune thyroid diseases requires the execution of additional randomized, double-blind, placebo-controlled trials and longer follow-up periods.

## 1. Introduction

Hashimoto’s thyroiditis (HT), also known as chronic autoimmune thyroiditis, is the most prevalent organ-specific autoimmune disorder whose frequency has increased considerably in recent decades. At present, HT constitutes one of the most common thyroid diseases, with an incidence of 0.3–1.5 cases per 1000 persons, especially in the female gender. HT is currently the leading cause of primary hypothyroidism, both in adolescents and adults. The main feature of the disease is the presence of thyroid autoantibodies against thyroid peroxidase (TPOAb) or thyroglobulin (TGAb). Antibody titers show a positive correlation with hypothyroidism. This condition is a T-cell-mediated autoimmune disorder characterized by thyroid lymphocytic infiltration. Even though the concrete etiology has not been fully elucidated, the pathogenesis of HT is thought to be related to the interaction among genetic influences, environmental triggers, and epigenetic effects [1,2].

The classic function of active vitamin D (calcitriol or 1,25-dihydroxyvitamin D3) is the regulation of calcium and phosphate concentrations, but recent evidence has suggested that vitamin D is also associated with non-skeletal roles [3,4]. Most body cells and organs (muscle, heart, blood vessels, pancreas, brain, mammary gland, colon, prostate, gonads, skin, immune cells, malignant cells, etc.) have nuclear vitamin D receptors (VDR) and activating enzymes for calcitriol synthesis that, in these locations, are not regulated by parathyroid hormone. Vitamin D performs most of its biological actions by binding to the VDR and, consequently, modulates the expression/transcription of numerous coding genes responsible for the regulation of cell proliferation, differentiation, and apoptosis (genomic pathway). Thus, vitamin D has pleiotropic effects and can even act in a paracrine or autocrine manner in addition to its endocrine function. This fact would explain the additional non-calciotropic effects of vitamin D, as its involvement in autoimmunity, endocrine, infectious, metabolic and neurological diseases, and mood disorders. Obviously, the expression of VDR in immune cells suggests that vitamin D plays a critical role in regulating both innate and adaptive immune systems. That is, vitamin D deficiency could compromise the integrity of the immune system and lead to inappropriate immune responses such as autoimmune diseases [5,6,7,8].

According to the US Endocrine Society’s guidelines, calcidiol levels are considered the best indicator of organic vitamin D content, given its long half-life (two to three weeks). Additionally, calcidiol concentrations below 20 ng/mL (<50 nmol/L) are considered to indicate vitamin D deficiency, whereas levels between 20 and 29 ng/mL (50–75 nmol/L) indicate a relative insufficiency, and levels of 30 ng/mL or greater indicate sufficient vitamin D. That is, optimal vitamin D levels range between 30 and 50 ng/mL (75–125 nmol/L) and maximum safe levels go up to 100 ng/mL (250 nmol/L) [9]. Although these cut-off points are based on observational studies, they are being accepted and used by most authors.

While there is current evidence that low vitamin D levels are a risk factor for autoimmune diseases (diabetes, multiple sclerosis, systemic lupus erythematosus, juvenile idiopathic arthritis, etc.), it remains uncertain if vitamin D deficiency is a significant factor in the pathogenesis or functional consequences of autoimmune hypothyroidism [1,10,11,12].

The aim of this review is to elaborate a comprehensive literature review (narrative review) about (a) recent data on the possible influence of vitamin D deficiency on the pathogenic mechanism of autoimmune hypothyroidism, (b) the relationship between vitamin D status and thyroid function and (c) the evaluation of the effect of vitamin D supplementation on thyroid function This review is based on an electronic search of literature performed by two independent researchers in the PubMed database of the U.S. National Library of Medicine published between January 2011 and December 2023. The following specific keywords (Medical Subject Headings) were used alone or in combination for the search: vitamin D or vitamin D deficiency/insufficiency, Hashimoto thyroiditis, autoimmunity, thyroid autoimmunity, and anti-thyroid antibodies (anti-thyroglobulin antibodies and anti-thyroid peroxidase).

## 2. Immunomodulatory Function of Vitamin D

Active vitamin D exerts most of its effects via the nuclear receptor VDR, a receptor that subsequently dimerizes with the “retinoid X receptor”; this heterodimer acts on the “vitamin D response element” and controls the expression/transcription of numerous coding genes responsible for the regulation of cellular proliferation, differentiation, and apoptosis (genomic pathway). The majority of immune cells, including T cells and B cells, and antigen-presenting cells (such as dendritic cells and macrophages), express the VDR and the vitamin D activating enzyme 1-a-hydroxylase. Thus, vitamin D status has the potential to directly condition the activation, proliferation and expression of the phenotype of immune cells, which, in turn, can directly activate vitamin D precursor (cells of the immune system respond to vitamin D and activate vitamin D in paracrine or autocrine ways). The wide expression of VDR in immune cells suggests that vitamin D plays a critical role in regulating both innate and adaptive immune systems. This classification (innate and adaptive immune systems) is merely didactic because both systems operate in an integrated way. The innate immune system activates the adaptive immune system, and the adaptive immune system uses the effector mechanisms of innate immunity to eliminate non-self-progenic antigenic elements.

### 2.1. Vitamin D and Innate Immunity

Vitamin D is an important mediator of innate immune responses (Figure 1). For example, vitamin D stimulates the differentiation of monocytes into macrophages and, consequently, improves the chemotaxis and phagocytic capabilities of innate immune cells. Likewise, vitamin D improves the antimicrobial properties of the immune cells (monocytes, neutrophils, and natural killer cells) and barrier epithelia through the modulation of gene expression of potent antimicrobial peptides (AMPs), such as cathelicidin and β-defensine, which destroy the cell membranes of bacteria and viruses. On the other hand, vitamin D has important effects on the natural killer cells: monocytes and dendritic cells (DCs). It increases the production of anti-inflammatory cytokines (IL-4 and IL-10) as well as inhibits the production of inflammatory cytokines such as interleukin IL-1, IL-2, IL-6, IL-8, IL-12, and interferon gamma (IFN-γ) and Toll-like receptors (TLR2 and TLR4), resulting in a stage of insufficient immune responsiveness. In this way, it helps avoid excessive innate responses and consequent tissue damage (systemic inflammation and/or septic shock). Additionally, it impairs DCs differentiation and maturation as evidenced by a decreased expression of major histocompatibility complex class II molecules (MHC-II), co-stimulatory molecules (CD40, CD80, and CD86), and CD1a (type of non-classical antigen-suplading molecule). The ultimate effect of vitamin D with the preservation of the immature phenotype of DCs is a reduction in the number of antigen-presenting cells and activation of naïve T cells, thus contributing to an induction of a tolerogenic state. In this way, the interaction between immature DCs (tolerogenic phenotype) and T cells induces T cell anergy and apoptosis and, secondarily, promotes the synthesis of tolerogenic cytokines. In fact, the inhibition of DC differentiation and maturation is particularly important in the context of autoimmunity and the suppression of self-tolerance.

### 2.2. Vitamin D and Adaptive Immunity

Vitamin D also plays a role in the regulation of adaptive immunity (Figure 1). Vitamin D modulates the activation and differentiation of naïve CD4+ lymphocytes after the presentation of the antigen by the DCs in the lymph nodes, resulting in a shift from a T-helper 1 (Th1) to a Th2 phenotype. This change implies inhibition of inflammatory cytokine production (IL-2, IFN-γ, and tumor necrosis factor alfa (TNF-α) and an increased production of anti-inflammatory cytokines (IL-4, IL-5, and IL-10). Furthermore, vitamin D affects differentiation to Th17 phenotype, leading to a decrease in the production of inflammatory cytokines such as IL-17 and IL-21 (linked to organ-specific autoimmunity, inflammation, and tissue damage), and facilitates the induction of T regulatory cells (Tregs) with increased production of anti-inflammatory cytokines such as IL-10 and transforming growth factor beta (TGF-β). Treg cells are able to suppress the proliferation and production of inflammatory cytokines by CD4+ T cells as well as the proliferation of CD8+ (cytotoxic lymphocytes) and antigen-presenting cells. In this way, vitamin D would modulate cell-mediated immune responses and regulate the inflammatory activity of T cells and, consequently, have a significant role in preventing exaggerated or autoimmune responses.

With regard to B-lymphocyte regulation, vitamin D has an impact on B cell homeostasis in several ways (through the inhibition of follicular T helper cells). For example, it reduces naïve B cell activation and proliferation, induces apoptosis of activated B cells, and suppresses the differentiation of B cells into plasma cells. In addition, vitamin D also inhibits memory B cell generation and reduces the immunoglobulin synthesis (IgG and IgM) in activated B cells. This control on B cell activation and proliferation may be clinically important in autoimmune diseases as B cells producing autoreactive antibodies play a major role in the pathophysiology of autoimmunity.

In, summary, vitamin D directly and indirectly influences and regulates both innate and adaptive immune cells since these cells widely express the VDR. In fact, vitamin D exerts an immunomodulatory role both in the innate and adaptive immune systems that appears to promote immune tolerance and therefore acts to decrease the likelihood of developing autoimmune disease [5,6,10,12,13,14,15,16,17].

## 3. Hashimoto’s Thyroiditis

Chronic autoimmune thyroiditis is characterized by self-tissue destruction via the adaptive immune responses that evade immune regulation. Under normal conditions, once the human body has obtained the ability of tolerance to certain antigens, the process of autoimmunity does not take place. However, whenever this process of tolerance is broken, autoimmunity occurs, just as it does in HT. It is characterized by a diffuse goiter, circulating anti-thyroid peroxidase (TPOAb) and/or anti-thyroglobulin (TGAb) antibodies, erratic degree of thyroid hypofunction, and intrathyroidal infiltration of B and T lymphocytes, with CD4+ type 1 T helper (Th1) subtype predominance. The antibody titer levels are positively correlated with the severity of thyroid inflammation and hypothyroidism. In particular, TPOAb is the most important autoantigen involved in the induction of autoimmune thyroid disease. Thyroid peroxidase has an essential role in the production of thyroid hormones (thyroxine and triiodothyronine) while thyroglobulin produces the storage of thyroid hormones in the thyroid follicles. The diagnosis of HT is based on clinical symptoms, anti-thyroid antibodies, and histological features. TPOAb is perceived as the most important feature of HT and is present in about 95% of patients. In contrast, TGAb is present in a lower (60–80%) percentage of cases and, therefore, these antibodies are less reliable for diagnosis. All these facts lead us to consider that TGAb may represent the expression of an initial immune response, whereas TPOAb may be the result of a posterior immune response in a way that simulates an immune escalation. At present, HT remains an incurable disease with an unpredictable evolution, often leading to lymphocytic destruction of the thyroid parenchyma that subsequently originates hypothyroidism and the need for thyroid hormone replacement for life [2].

The etiology of HT is multifactorial, involving (a) genetic predisposition (the role of genetic factors in HT pathogenesis is suggested both by the high concordance rate for HT in monozygotic twins and by the frequent finding of thyroid autoantibodies or other autoimmune diseases in blood relatives of the HT probands; the results of association studies of VDR polymorphisms -ApaI, BsmI TaqI, and Fok1- with autoimmune thyroid diseases are inconclusive), (b) environmental factors (e.g., radiation, infections, iodine, selenium intake, smoking, and dietary habits), and (c) endogenous factors (e.g., body mass index, adipokines, estrogens, microchimerism, glucocorticoids, and potentially the gastrointestinal microbiome). Environmental factors are thought to play an important role, as recent epidemiological changes have demonstrated, and the development of HT may be ascribed not only to innate predisposition but also to environmental factors that have changed rapidly. These factors might finally trigger the development of autoimmunity. Thereby, in those individuals with genetic predisposition, the disruption of these immune-endocrine interactions by environmental factors is the key to switching the physiological balance between the Th1 and Th2 immune response. This maladjustment results in a Th1-cell-mediated autoimmune reaction with thyrocyte destruction and secondary hypothyroidism in HT. Furthermore, a shift in the balance between Th17 and Treg cells in thyroid autoimmunity has been recently observed [15,18,19,20].

## 4. Autoimmune Mechanism of Autoimmune Hypothyroidism

The core of the autoimmune process in HT is a breakdown in self-tolerance to thyroid autoantigens that results in thyroid destruction by the infiltration of CD4+ Th1 cells, macrophages, and plasma cells. Additionally, these cells produce autoantibodies against thyroid peroxidase (TPOAb) and thyroglobulin (TGAb). Therefore, the detection of elevated titers of these antibodies is generally used for HT diagnosis [17].

First, the presence of environmental/genetic factors originates the activation of antigen-presenting cells (APCs), mainly DCs, which, in turn, present autoantigens to naive CD4+ T cells in lymph nodes. Consequently, these cells differentiate into Th1, Th2, Th17, or Tregs. As a remarkable fact, the follicular thyroid cells of HT patients may express MHC-II, which is crucial for presenting antigens to CD4+ T cells. Therefore, thyroid cells can act as APCs by presenting autoantigens to T cells and activating their differentiation. Second, the cytokines produced and released by Th1, including IL-2 and IFN-γ, induce the expression of MHC-II on the surface of the thyroid cells, and secondarily cause the differentiation of the naive CD4+ T cells into Th1. Finally, Th1 cells, via IL-2 and IFN-γ, induce the activation of CD8+ T cells (cytotoxic T cells). CD8+ T cells induce apoptosis of thyroid cells (Figure 2), which leads to the release of pro-inflammatory cytokines that contribute to the activity and migration of pathological Th17 cells and the suppression of Tregs cells, and consequently amplifies and sustains the immune feedback process. Some authors have suggested that Th1/Th2 cell imbalance and Th1 cell activity enhancement would be the main contributors to the development of HT, but other authors estimate that increased Th17/Treg ratio would play a critical role in the pathogenesis of autoimmune thyroid diseases. This results in positive feedback and the initiation of the autoimmune process and subsequent consolidation. However, in the destruction of the thyroid gland and the mechanism of the autoimmune process, a humoral response also occurs.

The recruitment of Th1 lymphocytes is associated with the stimulation of B lymphocytes, which are located in lymph nodes within the tissue of the thyroid gland. Infiltrating B cells release autoantibodies, mainly TPOAb and TGAb, which are thyroid self-antigens. These antibodies contribute to the apoptosis of thyroid follicular cells in the mechanism of antibody-dependent cell-mediated cytotoxicity [2,8,14,17,21,22]. In the course of HT, self-reactive CD4+ T lymphocytes recruit B cells and CD8+ T cells that gather into the thyroid gland. Finally, the progression of the disease leads to the death of thyroid cells and hypothyroidism (Figure 2).

## 5. Immunomodulatory Role of Vitamin D in Hashimoto’s Thyroiditis

As described above, vitamin D exerts an immunomodulatory role both in the innate and adaptive immune systems that appears to promote immune tolerance, and, in this case, it could contribute to the inhibition of the immunopathological process in HT. On one hand, active vitamin D promotes the differentiation of monocytes into macrophages and stimulates antimicrobial activity via a series of mechanisms that increase the transcription of antimicrobial peptide genes, such as beta-defensin and cathelicidin antimicrobial peptide. That is, vitamin D improves chemotaxis and phagocytic capabilities and antimicrobial properties of innate immune cells. On the other hand, vitamin D also plays a role in the regulation of adaptive immunity, as it modulates the activation and differentiation of naïve CD4+ lymphocytes after antigen presentation by DCs in lymph nodes. In other words, Vitamin D carries out an immunomodulatory role both in the innate and adaptive immune systems that appears to promote immune tolerance, and, in this case, it could contribute to the inhibition of the immunopathological process in HT.

The potential mechanisms by which vitamin D could contribute to inhibiting the autoimmune process in HT would be varied and complementary and could be summarized as follows:

(a)Vitamin D inhibits the expression of various proinflammatory cytokines from DCs (IL-2, IL-6, and IL-12) that activate T cells while enhancing the expression of IL-10 (anti-inflammatory or tolerogenic cytokine); this results in a stage of insufficient immune responsiveness and, in this way, it helps avoid excessive innate responses and consequent tissue damage (systemic inflammation and/or septic shock). Additionally, vitamin D impairs DCs differentiation and maturation as evidenced by a decreased expression of MHC-II and co-stimulatory molecules (CD40, CD80, and CD86); this preservation of the immature phenotype of DCs results in a reduction in the number of antigen-presenting cells and activation of naïve T cells, thus contributing to an induction of a tolerogenic state. Vitamin D also modulates the activation and differentiation of naïve CD4+ lymphocytes after the presentation of the antigen by the DCs in the lymph nodes, resulting in a shift from a T-helper (Th)1 to a Th2 phenotype, which is an inhibition of inflammatory cytokine production (IL-2, IFN-γ, and TFN-α), and an increased production of anti-inflammatory cytokines (IL-4, IL-5, and IL-10).(b)Vitamin D may reduce MHC-II expression in the follicular thyroid cells, thus preventing T cell activation and proinflammatory cytokine response.(c)Vitamin D affects the differentiation of naïve T cells towards the Th17 phenotype, leading to a decrease in the production of inflammatory cytokines such as IL-17 (linked to organ-specific autoimmunity, inflammation, and tissue damage), and facilitates the induction of T regulatory cells (Tregs) with increased production of anti-inflammatory cytokines such as IL-10 and TGF-β. Treg cells are able to suppress the proliferation and production of inflammatory cytokines by CD4+ T cells as well as the proliferation of CD8+ (cytotoxic lymphocytes) and APCs. Therefore, vitamin D contributes to the restoration of the Th17/Treg ratio (Th17 cells mainly express proinflammatory activity, which secondarily causes the development of autoimmune disorders; Tregs modulate the immune system and maintain tolerance to self-antigens, which in turn prevents autoimmunity). In this way, vitamin D would modulate cell-mediated immune responses and regulate the inflammatory activity of T cells and, consequently, have a significant role in preventing exaggerated or autoimmune responses.(d)Finally, with regard to B-lymphocyte regulation, vitamin D has an impact on B cell homeostasis in several ways. For example, it reduces naïve B cell activation and proliferation, induces apoptosis of activated B cells as well as suppresses the differentiation of B cells into plasma cells. In addition, vitamin D also inhibits memory B cell generation and reduces immunoglobulin synthesis (IgG and IgM). This control on B cell activation and proliferation may be clinically important in HT, as B cells producing autoreactive antibodies play a major role in the pathophysiology of autoimmunity.

Thereby, vitamin D induces a shift from a pro-inflammatory to a more tolerogenic immune status, resulting in a limitation in the development of self-reactive T cells preventing inflammation and autoimmunity [6,7,10,17,22,23]. Thus, vitamin D appears to play an important immunomodulatory role, and its relationship with autoimmune thyroid disease has been widely studied in recent years.

In addition, a high Th22 cell count has been reported in the blood and thyroid of HT patients, considered decisive in the inflammatory effects of IL-22 on thyroid cells [18].

## 6. Relationship between Vitamin D Status and Hashimoto’s Thyroiditis

Many observational studies (case–control or cross-sectional studies) have unveiled a potential link between hypovitaminosis D and an increased risk of HT onset [23,24,25,26,27,28,29,30,31,32,33,34,35]. In fact, considering the adult population, a low vitamin D status has been reported in patients with autoimmune thyroid diseases or HT, suggesting an association between vitamin D deficiency and thyroid autoimmunity. Furthermore, several authors reported that the prevalence of vitamin D deficiency in patients with HT was significantly higher compared with healthy individuals, and that serum calcidiol levels were inversely correlated with anti-thyroid antibodies (TGAb and TPOAb), suggesting the involvement of vitamin D in its pathogenesis. In addition, considering all the HT cases, patients with hypothyroidism showed a higher prevalence of vitamin D deficiency and lower calcidiol levels in comparison to patients with euthyroidism or healthy individuals.

On the other hand, few studies have focused on investigating the potential correlation between low vitamin D levels and HT in children, whose reported results are similar to those of adults. That is, these authors conclude that low serum vitamin D levels are significantly associated with autoimmune thyroid diseases or HT, also observed in children, although they also indicate that it is not an independent risk factor for the progression to overt hypothyroidism [36,37,38,39,40].

However, the results of other observational studies have not found a relationship between vitamin D levels and antithyroid antibodies or thyroid function [19,41,42,43,44]. Different factors could contribute to this discordance between the studies. These include the application of different commercially available kits for serum vitamin D assay or the consensus on a definition of vitamin D deficiency between the studies, as well as potential confounding factors such as BMI and ethnic, seasonal, or geographical differences [18].

Nevertheless, recent systematic reviews, meta-analyses, and meta-regression of observational studies [44,45,46,47,48,49] have confirmed that vitamin D levels were significantly lower in autoimmune hypothyroidism disease or HT patients compared to healthy people. Accordingly, it seems that there is proven evidence supporting a relationship between low vitamin D status and HT.

Further, several authors have reported an inverse correlation between thyrotrophin (TSH) and Vitamin D status in healthy young people as well as in middle-aged and older men [50,51]. In addition to its association with autoimmune thyroid diseases, vitamin D deficiency has also been detected in other autoimmune diseases, such as multiple sclerosis, diabetes mellitus, systemic lupus erythematosus, and others [11,12]

## 7. Vitamin D Supplementation in Hashimoto’s Thyroiditis

The large amount of data suggesting an association between low levels of vitamin D and autoimmune thyroid diseases has raised concerns and boosted the investigation of the use of vitamin D supplements (cholecalciferol) in the prevention or treatment of HT. At present, there is still no specific drug to reduce antibody titers, but sufficient evidence linking vitamin D deficiency to HT has led to clinical trials that intend to examine the potential beneficial effects of vitamin D supplementation in the prevention and treatment of autoimmune thyroid disorders.

Obviously, the effects of the immune intervention of vitamin D supplementation on experimental autoimmune thyroiditis had previously been explored. Several authors reported that thyroid autoantibodies and IFN-γ and IL-12 levels (pro-inflammatory cytokines) decreased significantly, but IL-4 and IL-10 levels (anti-inflammatory cytokines) increased markedly with vitamin D supplementation in female Wistar rats with experimental autoimmune thyroiditis. That is to say, vitamin D seemed to regulate the production of thyroid autoantibodies and the imbalance of cytokines in animal models [52,53].

Most of the available randomized controlled trials have been carried out in the adult population and have found that, after supplementation with cholecalciferol in patients diagnosed with HT and vitamin D deficiency, thyroid autoantibodies (TPOAb and TgAb) titers decreased significantly, posing that vitamin D treatment may have a beneficial effect on autoimmunity hypothyroidism (in these studies, calcidiol concentration increased, reaching above 30 ng/mL). In addition, a reduction in antibody levels was also observed in subjects with normal baseline vitamin D status [54].

In fact, in the recent decade, several studies have demonstrated a significant reduction in TPOAb and/or TgAb in adult patients diagnosed with autoimmune thyroiditis after vitamin D supplementation in different populations (Table 1): Greece [29], India [55,56], Turkey [57], Iran [58,59], Poland [54,60], and Canada [61]. The dose used in vitamin D supplementation has been highly variable, ranging from 1000 to 4000 IU/daily for 1–6 months [29,54,57,60]; or with weekly doses between 50,000 and 60,000 IU for 2–3 months [55,56,58,59]. In addition, some researchers have recently reported that vitamin D supplementation may influence other immunological markers. Supplementation with 50.00 IU of vitamin D weekly for 3 months in adult patients with TH resulted in beneficial immunological effects through a significant decrease in the Th17/Treg ratio and inflammatory factors, such as IFN-γ and TNF-α, as well as a significant decrease in IL-10 (pro-inflammatory cytokine) [59,62]. The wide variety of both doses and duration of vitamin D supplementation makes it risky to establish and, consequently, recommend an optimal vitamin D regimen in patients with HT.

However, a recent meta-analysis of randomized controlled trials concludes that most of these studies have not detected any significant change in TSH levels after vitamin D supplementation; additionally, thyroid hormones remained mostly unchanged after vitamin D supplementation [11]. Even though numerous studies concerning antithyroid antibodies after vitamin D supplementation gave consistent outcomes, the fact that some investigations showed contrary results should be mentioned [63,64,65]. Nevertheless, recent systematic reviews and meta-analyses have confirmed that vitamin D supplementation is associated with a reduction in TGAb and TPOAb titers in patients with HT in the short-term. The effect was more intense if patients received cholecalciferol and the duration of treatment was more than 3 months [66,67], However, there are no conclusive results on the effect of vitamin D supplementation on the thyroid function.

The results found in a study with a large number of patients with HT (case–control design) are particularly remarkable. Participants received doses of vitamin D (between 1000 and 4000 IU daily) to achieve optimal concentrations (>40 ng/mL). After a period of 12 months of follow-up, a significant reduction in antithyroid antibodies was observed. Furthermore, it was found that a concentration of calcidiol >50 ng/mL was associated with a substantial improvement in thyroid function, including a reduction in TSH levels and symptom severity. That is, vitamin D would offer a safe and economical approach to improving thyroid function and could provide protection against the development of autoimmune thyroiditis [61].

Exceptionally, clinical trials have been carried out in pediatric age. A recent randomized clinical trial performed exclusively in pediatric patients with HT (average age: 12.6 years) showed that the level of autoantibodies and thyroid volume decreased after vitamin D supplementation. Patients with hypovitaminosis (<30 ng/mL) received 50,000 IU pearl every week for 6 weeks; subsequently, the serum calcidiol level was rechecked, and if it reached the normal range (>30 ng/mL), the patients continued the intake of cholecalciferol once a month for 6 months. Antithyroid antibody titers (TPOAb and TGAb) decreased significantly, as did the size of both thyroid lobes after intervention [68]. Thus, it is suggested that serum vitamin D level should be routinely checked in these patients and, when observing hypovitaminosis, an appropriate treatment with vitamin D should be carried out.

## 8. Conclusions

Vitamin D carries out an immunomodulatory role both in the innate and adaptive immune systems that appears to promote immune tolerance, and, in this case, it could contribute to the inhibition of the immunopathological process in HT. Although the mechanisms underlying the interrelationship of vitamin D and HT remain unknown, they are likely related to their anti-inflammatory and immunomodulatory properties. There is extensive literature studying the complex relationship between vitamin D deficiency and autoimmune thyroid diseases; however, data are inconclusive. The need to resolve the ambiguity concerning the causal relationship between vitamin D and thyroid disorders turns out to be essential, as vitamin D supplementation is inexpensive and has minimal side effects, and this would make it potentially revolutionary in the treatment of hypothyroidism.

There is still a gap in the knowledge regarding the potential of vitamin D supplementation in the treatment of HT patients. There is a need to clarify if vitamin D supplementation is helpful in decreasing the replacement dose of levothyroxine or if it will stop the need for levothyroxine replacement if used in the early stages of HT.

The confirmation of the beneficial effects of vitamin D in the prevention and treatment of autoimmune thyroid diseases requires additional randomized, double-blind, placebo-controlled trials with longer periods of follow-up. However, some authors have suggested that, when diagnosing patients with HT, vitamin D replacement may be suggested as an auxiliary treatment, and they even predict that, in the future, vitamin D could become part of the treatment of autoimmune thyroid diseases, especially in those individuals with vitamin D insufficiency.

## Figures and Tables

**Figure 1 ijms-25-03154-f001:**
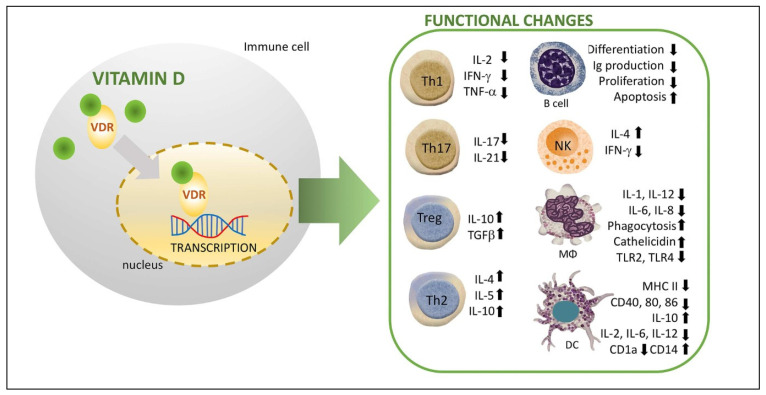
Scheme of immunomodulatory effects of active vitamin D on various immune cell lineages (adapted from [6]). Effects of vitamin D on both NK, Mφ, and DCs include inhibition of inflammatory cytokine production and inhibition of DCs differentiation and maturation, which in turn leads to suppression of T cell proliferation and results in a shift from a Th1 to a Th2 phenotype. Vitamin D affects T cell maturation by skewing away from the inflammatory Th17 phenotype and facilitates the induction of Treg cells. In addition, vitamin D reduces B cell activation and proliferation and immunoglobulin synthesis and induces apoptosis of activated B cells. DC: dendritic cell. Mφ: macrophage. NK: natural killer. VDR: vitamin D receptor.

**Figure 2 ijms-25-03154-f002:**
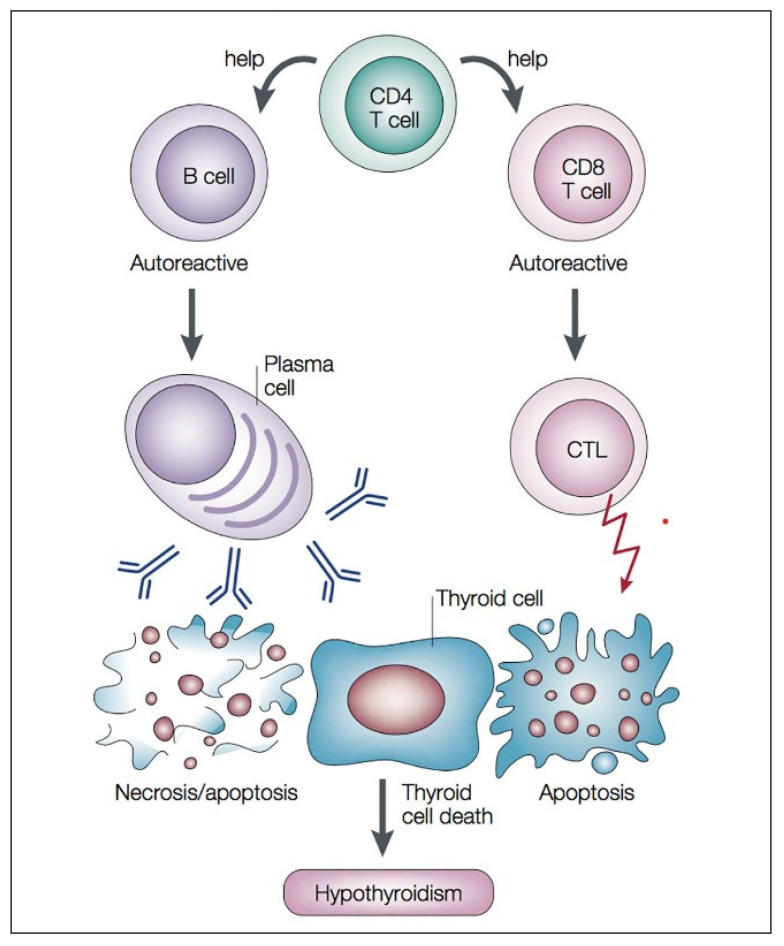
Autoimmune process: pathogenic necrosis/apoptosis mechanism of thyroid follicular cells (adapted from [2]). In the course of HT, CD4+ T lymphocytes recruit B cells and CD8+ T cells into the thyroid. Finally, the progression of the disease leads to the death of thyroid cells and hypothyroidism. CTL: cytotoxic T lymphocytes.

**Table 1 ijms-25-03154-t001:** Several studies with significant reduction in antithyroid antibody titers after vitamin D (cholecalciferol) supplementation.

Author, Year, and Country	Number of Participants (F/M)	Dose of Supplementation Duration
Mazokopakis et al., 2015 (Greece) [29]	173 F/13 M	1200–4000 IU/daily for 4 months
Chaudhary et al., 2016 (India) [55]	39 F/11 M	60,000 IU weekly, 8 weeks
Simsek et al., 2016 (Turkey) [57]	37 F/9M	1000 IU/daily for 1 month
Krysiak et al., 2017 (Poland) [54]	34 F	2000 IU daily, 6 months
Mirhosseini et al., 2017 (Canada) [61]	103	Doses modified to achieve calcidiol concentration >40 ng/mL, 12 months
Chahardoli et al., 2019 (Iran) [58]	42 F	50,000 IU weekly, 3 months
Krysiak et al., 2022 (Poland) [60]	42 F	4000 IU daily for 6 months
Bhakat et al., 2023 (India) [56]	50	60,000 IU weekly for 8 weeks
Robat-Jazi et al., 2022 (Iran) [59]	40 F	50,000 IU weekly, 3 months

F: females. M: males.

## Data Availability

Not applicable.

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
