# Peer review of "Autoimmune Thyroiditis and Vitamin D"

_ijms, 2024, doi:10.3390/ijms25063154_

Round 1

Reviewer 1 Report

Comments and Suggestions for Authors

The manuscript review by Dura-Trave summarizes the available evidence on the role of vitamin D in autoimmunity thyroiditis. It is a well-writen manuscript and summarizes most the research that has been done on this issue. They conclude that there is not conclusive evidence on the effect of vitamin D in this pathology. Although, this work sounds good, I have some comments:

1. Figures should be clearly explained. There are not figure leyends on both of them.

2. Are there any animal models on autoimmune thyroiditis? If yes, please include results described on those models.

Author Response

Reply to reviewer-1

Comments and Suggestions for Authors

The review manuscript by Dura-Trave et al. addresses whether the supplementation of vitamin D affects autoimmune thyroiditis. This manuscript is important in the field since it summarizes very well recent findings on this topic. They conclude that although vitamin D is significantly lower in patients with autoimmune thyroiditis, there is no conclusive evidence that supplementation with vitamin D could affect patients with autoimmune thyroiditis. The manuscript is well-written, the references are updated, and it includes the most relevant literature on the subject area. Most of the relevant issues have been addressed in the review and updated recent findings on this subject area. Their conclusions are consistent with the evidence presented in the manuscript. However, I have some specific comments, which should be addressed before publication.

First, we would like to thank you for your encouraging words regarding this article.

NOTE: The corrected text of the new version is in red

  1. A list of all used abbreviations should be included in the text. Many of them are lacking.

The following abbreviations have been included in the list of abbreviations:

Mφ: macrophage

NK : natural killer cells

TLR: toll-like receptors

  1. Legends of Figures 1 and 2 must be expanded and clearly described.

The legends of the figures have been enlarged for a better understanding

Where it says...

Figure 1. Immunomodulatory effects of active vitamin D on various immune cells lineages (adapted from 6).

You should read...

Figure 1. Scheme of immunomodulatory effects of active vitamin D on various immune cells lineages (adapted from 6). Effects of vitamin D on both NK, Mφ and DCs include inhibition of inflammatory cytokine production and inhibition of DCs differentiation and maturation, which in turn leads to suppression of T cell proliferation and results in a shift from a Th1 to a Th2 phenotype. Vitamin D affects T cell maturation with a skewing away from the inflammatory Th17 phenotype and facilitates the induction of Tregs cells. In addition, vitamin D reduces B cells activation and proliferation and immunoglobulin synthesis, and induces apoptosis of activated B cells

.

Where it says...

Figure 2. Autoimmune process: pathogenic necrosis/apoptosis mechanism of thyroid follicular cells. (adapted from 2).

You should read...

Figure 2. Autoimmune process: pathogenic necrosis/apoptosis mechanism of thyroid follicular cells (adapted from 2). In the course of HT, CD4+ T lymphocytes recruit B cells and CD8+ T cells into the thyroid. Finally, the progression of the disease leads to the death of thyroid cells and hypothyroidism.

  1. It is needed a more detailed description of the studies presented in Table I.

The following paragraph has been added

In fact, in the last decade, several studies have demonstrated a significant reduction in TPOAb and/or TgAb in adult patients diagnosed with autoimmune thyroiditis after vitamin D supplementation in different population: Greece (29), India (55, 56), Turkey (57), Iran (58, 59), Poland (54, 60) and Canada (61). The dose used in vitamin D supplementation has been highly variable, ranging from 1,000 to 4,000 IU/daily for 1-6 months (29, 54, 57, 60), or with weekly doses between 50,000 and 60,000 IU for 2-3 months (55, 56, 58, 59). In addition, some researchers have recently reported that vitamin D supplementation may influence other immunological markers. Supplementation with 50,000 IU weekly of vitamin D for 3 months in adult patients with TH resulted in beneficial immunological effects through a significant decrease in the Th17/Treg ratio and inflammatory factors, such as IFN and TNF, as well as a significant decrease in IL-10 (pro-inflammatory cytokine) (59, 62). The wide variety of both doses and duration of vitamin D supplementation makes it risky to establish and, consequently, recommend an optimal vitamin D regimen in patients with HT.

  1. Are there any animal models for autoimmune thyroiditis? If yes, please describe them and compare them with the findings in human patients.

The following paragraph has been added

Obviously, the effects of the immune intervention of vitamin D supplementation on experimental autoimmune thyroiditis had previously been explored. Several authors reported that thyroid autoantibodies and IFN-γ and IL-12 levels (pro-inflammatory cytokines) decreased significantly, but IL-4 and IL-10 levels (anti-inflammatory cytokines) increased markedly with vitamin D supplementation in female Wistar rats with experimental autoimmune thyroiditis. That is to say, vitamin D seemed to regulate the production of thyroid autoantibodies and the imbalance of cytokines in animal models (52, 53)

  1. Finally, a hypothesis or an explanation should be put forward to explain why vitamin D supplementation could not be effective in autoimmune thyroiditis. Also, prospects or directions in the field should be given to focus on the research subject

According to the US Endocrine Society´s guidelines the optimal vitamin D levels would range between 30-50 ng/mL (75-125 nmol/L) (see new paragraph in Introduction…). In fact, all the authors cited who have investigated the effects of the immune intervention of vitamin D supplementation on autoimmune thyroiditis (table 1), as stated in the text, have tried to achieve calcidiol levels within this range: “…(in these studies, calcidiol concentration increased reaching above 30 ng/ml)…”

.

However, as can be read in the Discussion (ref. 61, Mirhosseini, et al, 2017), a particularly remarkable study found, after 12 months with vitamin D supplementation, that, at the moment calcidiol concentration exceeded 50 ng/ml, not only a significant decrease in thyroid autoantibodies but also a substantial improvement in thyroid function (reduction in TSH levels and symptom severity) was observed.

Therefore, although the wide variety of doses and duration of vitamin D supplementation (table 1) makes it risky to establish and, consequently, recommend an optimal vitamin D regimen in patients with HT, it could be hypothesized that the low effectiveness of published trials of vitamin D supplementation in autoimmune thyroiditis could be due either to a short duration of supplementation or to the fact that its therapeutic effectiveness required calcidiol concentrations above 50 ng/ml. Current data are far from conclusive. In fact, practically all authors conclude their studies by indicating (as can be read in the conclusions) that more studies (preferably randomized, double-blind, placebo-controlled trials with longer follow-up periods) are needed to confirm the beneficial effects of vitamin D in the prevention and treatment of autoimmune thyroid diseases.

We would like to express our thanks to referee for your suggestions and positive criticisms.

We hope every made question have been answered adequately.

Yours sincerely,

Teodoro Durá-Travé

Reviewer 2 Report

Comments and Suggestions for Authors

Authors of the manuscript presents an interesting review on the relationship between vitamin D and thyroid function. Moreover, they discussed recent studies on the vitamin D status in patients with Hashimoto’s thyroiditis and the effect of vitamin D supplementation on thyroid function. Several issues need to be addressed before the manuscript acceptance.

1.     Introduction: line 37: “1,25-hydroxyvitamin D” should be corrected to “1,25-dihydroxyvitamin D3”

2.     Introduction, lines 54-57: Author should mention which vitamin D form is measured to determine the vitamin D body status. Moreover, they should provide current recommendations regarding normal vitamin D levels and values which are regarded as deficient.

3.     Line 237: correct interleukins (IL-12 is listed twice)

4.     Table 1: provide additional information, e.g. number of patients, gender, adults or children. Concentrations of calcidiol (>40) should be expressed in “ng/ml” instead of “mg/ml”. In the study of Nodehi et al., the information on changes in TPOAb and TGAb is missing.

5.     References 61-64 should be discussed in the text. 

6.     Based on the discussed results of different studies, could Authors recommend the optimal concentration of vitamin D in patients with Hashimoto’s disease and/or the dose of supplementation?

Author Response

Reply to reviewer-2

Comments and Suggestions for Authors

Authors of the manuscript presents an interesting review on the relationship between vitamin D and thyroid function. Moreover, they discussed recent studies on the vitamin D status in patients with Hashimoto’s thyroiditis and the effect of vitamin D supplementation on thyroid function. Several issues need to be addressed before the manuscript acceptance.

We would like to thank you for your encouraging words regarding this article.

NOTE: The corrected text of the new version is in red

  1. Introduction: line 37: “1,25-hydroxyvitamin D” should be corrected to “1,25-dihydroxyvitamin D3”

On line 37 (first version), "1,25-hydroxyvitamin D" has been corrected to "1,25-dihidroxivitamin D3)."

  1. Introduction, lines 54-57: Author should mention which vitamin D form is measured to determine the vitamin D body status. Moreover, they should provide current recommendations regarding normal vitamin D levels and values which are regarded as deficient.

The following paragraph has been added

According to the US Endocrine Society´s guidelines calcidiol levels are account as the best indicator of organic vitamin D content, given its long half-life (two to three weeks). Additionally, calcidiol concentrations below 20 ng/ml (< 50 nmol/L) are considered to indicate vitamin D deficiency, whereas levels between 20 and 29 ng/ml (50–75 nmol/L) indicate a relative insufficiency, and levels of 30 ng/ml or greater indicate sufficient vitamin D. That is, optimal vitamin D levels would range between 30-50 ng/mL (75-125 nmol/L) and maximum safe levels up to 100 ng/mL (250 nmol/L) (9). These cut-off points, although based on observational studies, are being accepted and used by most authors.

  1. Line 237: correct interleukins (IL-12 is listed twice)

On line 45 (first version) there is a semantic error:

Where it says... (IL-12, IL-6 and IL-12)

You should read... (IL-2, IL-6 and IL-12)

  1. Table 1: provide additional information, e.g. number of patients, gender, adults or children. Concentrations of calcidiol (>40) should be expressed in “ng/ml” instead of “mg/ml”. In the study of Nodehi et al., the information on changes in TPOAb and TGAb is missing.

Table 1 has been modified/corrected:

     -The number of participants and gender (female & male) have been added,

     -In this new version, the text clarifies that all patients were adults:

…Most of the available randomized controlled trials have been carried out in the adult population and have found that….

…in the last decade, several studies have demonstrated a significant reduction in TPOAb and/or TgAb in adult patients diagnosed with autoimmune thyroiditis… (see answer to the next question…)

In addition, as can be read in the last paragraph of the Discussion: “Exceptionally, clinical trials have been carried out in pediatric age…” (reference 00, Aghili et al, 2020). The results were very similar to those of adults

Table 1. Several studies with significant reduction in antithyroid antibody titers after vitamin D (cholecalciferol) supplementation.

Author, year and country

Number of participants (F/M)

Dose of supplementation Duration

Mazokopakis et al, 2015 (Greece) (29)

173F /13M

1200-4000 IU/daily for 4 months

Chaudhary et al, 2016 (India) (55)

39F/11M

60,000 IU weekly, 8 weeks

Simsek et al, 2016 (Turkey) (57)

37 F/9M

1000 IU/daily for 1 month

Krysiak et al, 2017 (Poland) (54)

34 F

2000 IU daily, 6 months

Mirhosseini et al, 2017 (Canada) (61)

103

Doses modified to achieve calcidiol concentration >40 ng/ml, 12 months

Chahardoli et al, 2019 (Iran) (58)

42 F

50,000 IU weekly, 3 months

Krysiak et al, 2022 (Poland) (60)

42 F

4000 IU daily for 6 months

Bhakat et al, 2023 (India) (56)

50

60,000 IU weekly for 8 weeks

Robat-Jazi et al. 2022 (Iran) (59)

40 F

50,000 IU weekly, 3 months

  1. References 61-64 should be discussed in the text. 

The following paragraph has been added

In fact, in the last decade, several studies have demonstrated a significant reduction in TPOAb and/or TgAb in adult patients diagnosed with autoimmune thyroiditis after vitamin D supplementation in different population: Greece (29), India (55, 56), Turkey (57), Iran (58, 59), Poland (54, 60) and Canada (61). The dose used in vitamin D supplementation has been highly variable, ranging from 1,000 to 4,000 IU/daily for 1-6 months (29, 54, 57, 60), or with weekly doses between 50,000 and 60,000 IU for 2-3 months (55, 56, 58, 59). In addition, some researchers have recently reported that vitamin D supplementation may influence other immunological markers. Supplementation with 50,000 IU weekly of vitamin D for 3 months in adult patients with TH resulted in beneficial immunological effects through a significant decrease in the Th17/Treg ratio and inflammatory factors, such as IFN and TNF, as well as a significant decrease in IL-10 (pro-inflammatory cytokine) (59, 62). The wide variety of both doses and duration of vitamin D supplementation makes it risky to establish and, consequently, recommend an optimal vitamin D regimen in patients with HT.

  1. Based on the discussed results of different studies, could Authors recommend the optimal concentration of vitamin D in patients with Hashimoto’s disease and/or the dose of supplementation?
  2. a) According to the US Endocrine Society´s guidelines the optimal vitamin D levels would range between 30-50 ng/mL (75-125 nmol/L) (see answer to the second question…). In fact, all the authors cited who have investigated the effects of the immune intervention of vitamin D supplementation on autoimmune thyroiditis (table 1), as stated in the text, have tried to achieve calcidiol levels within this range: “…(in these studies, calcidiol concentration increased reaching above 30 ng/ml)…”

.

However, the text refers to a particularly remarkable study corresponding to reference 61 (Mirhosseini, et al, 2017):

“The results found in a study with a large number of patients with HT (case-control design) are particularly remarkable. Participants received doses of vitamin D (between 1,000 and 4,000 IU daily) to achieve optimal concentrations (>40 ng/ml). After 12 months of follow-up, a significant reduction in antithyroid antibody titer was observed. Furthermore, it was found that concentration of calcidiol >50 ng/ml was associated with a substantial improvement in thyroid function, including a reduction in TSH levels and symptom severity…”

  1. b) The following paragraph has been added:

The wide variety of both doses and duration of vitamin D supplementation makes it risky to establish and, consequently, recommend an optimal vitamin D regimen in patients with HT.

In fact, practically all authors conclude their studies by indicating (as can be read in the conclusions) that more studies (preferably randomized, double-blind, placebo-controlled trials with longer follow-up periods) are needed to confirm the beneficial effects of vitamin D in the prevention and treatment of autoimmune thyroid diseases.

We would like to express our thanks to referee for your suggestions and positive criticisms.

We hope every made question have been answered adequately.

Yours sincerely,

Teodoro Durá-Travé